# Screening for Diabetes Complications during the COVID-19 Outbreak in South Korea

**DOI:** 10.3390/ijerph19095436

**Published:** 2022-04-29

**Authors:** Yu shin Park, Soo Young Kim, Eun-Cheol Park, Sung-In Jang

**Affiliations:** 1Department of Public Health, Graduate School, Yonsei University, Seoul 03772, Korea; dbtls0459@yuhs.ac (Y.s.P.); skim21@yuhs.ac (S.Y.K.); 2Institute of Health Services Research, Yonsei University, Seoul 03772, Korea; ecpark@yuhs.ac; 3Department of Preventive Medicine, Yonsei University College of Medicine, Seoul 03722, Korea

**Keywords:** COVID-19, diabetes, diabetic retinopathy, diabetic nephropathy

## Abstract

This study aimed to investigate the implementation of diabetes complications screening in South Korea during the coronavirus disease (COVID-19) outbreak. Data from the Korea Community Health Surveys conducted in 2019 and 2020 were used. This study included 51,471 participants. Multiple level analysis was used to investigate the relationships between screening for diabetic retinopathy and diabetic nephropathy and variables of both individual- and community-level factors in 2019 and 2020, before and after the COVID-19 outbreak. Diabetes nephropathy complications screening in 2020 had a lower odds ratio. However, regions heavily affected by COVID-19 showed a negative association with diabetes complications screening after the COVID-19 outbreak. For those being treated with medication for diabetes, there was a significant negative association with diabetic nephropathy screening after the outbreak. The COVID-19 outbreak was associated with a reduction in the use of diabetes nephropathy complications screening. Additionally, only regions heavily affected by COVID-19 spread showed a negative association with diabetes complications screening compared to before the COVID-19 outbreak. In this regard, it appears that many patients were unable to attend outpatient care due to COVID-19. As such, these patients should be encouraged to visit clinics for diabetes complications screening. Furthermore, alternative methods need to be developed to support these patients. Through these efforts, the development of diabetes-related complications should be prevented, and the costs associated with these complications will be reduced.

## 1. Introduction

An estimated 415 million adults worldwide have diabetes. The prevalence of diabetes in adults over the age of 18 years in OECD countries was 8.3% in 2019 [1], and the prevalence of diabetes among adults over the age of 30 years in South Korea was 14.5% in 2019 [2]. The prevalence of diabetes worldwide is expected to reach 10.4% by 2040 [3]. Moreover, global medical expenses associated with diabetes was estimated to total at least USD 376 billion in 2010 and expected to spend USD 893 billion in 2030 [4]. Both the prevalence and economic burden of diabetes are expected to substantially increase in the future [3].

Diabetes and its complications reduce the quality of life and worsen the health condition of patients. Diabetic retinopathy (DR), a principal complication of diabetes, is a leading cause of preventable blindness and visual impairment [5]. The prevalence of DR among patients with diabetes is 34.6% globally [6]. Diabetic nephropathy (DN), one of the most frequent complications of diabetes, is a major cause of end-stage renal disease [7]. Progression to microalbuminuria develops in 2.0% of patients in the first year, and approximately 25% of patients with diabetes develop microalbuminuria or more severe nephropathy within 10 years of diagnosis [8]. Despite the low healthcare costs for early treatment of retinopathy and microalbuminuria, these conditions can progress to more costly advanced diseases if left untreated [9,10]. Comprehensive eye examinations and urine albumin excretion tests are recommended annually for all patients with type 2 diabetes [11].

Coronavirus disease (COVID-19) caused a global pandemic, with the World Health Organization (WHO) declaring a public health emergency internationally in 2020 [12]. Since the first case of COVID-19 infection was identified in South Korea on 20 January 2020, the first large outbreak occurred in Daegu, a city of 2.5 million people [13]. This precipitated the first wave of COVID-19 in South Korea [14]. For the medical system of Daegu, one of the earliest consequences of the surge in COVID-19 cases was a shortage of hospital beds, supplies, and healthcare workers [15].

One report showed that the total healthcare claimed for health insurance in South Korea decreased compared to that in the previous year despite an increase in COVID-19 care and examinations [16]. Even during the 2015 Middle East respiratory syndrome (MERS) outbreak in Korea, it was found that outpatients at all levels of hospitals significantly decreased. The number of outpatients declined significantly in areas where many MERS confirmed cases occurred [17]. In the USA, outpatient visits for health check-ups decreased significantly during the COVID-19 pandemic because they were substantially affected by closure policies [18,19]. In addition, the total numbers of pediatric admissions and visits to emergency departments decreased after lockdowns globally [20,21].

Since then, Korea has experienced several waves of COVID-19 infections. Health centers and other hospitals have focused their efforts and attention on the diagnosis and treatment of patients with COVID-19 in the community, subsequently reducing or suspending the care of general patients. Research on this response to COVID-19 is necessary in order to understand the collateral effects on the prevention and treatment of other conditions. To the best of our knowledge, no study has examined the effects on screening for diabetes complications during the COVID-19 pandemic. The aim of this study was to investigate the differential effects on diabetes complications screening before and after the pandemic and, in particular, examine the factors involved.

## 2. Materials and Methods

For individual-level factors, we used data from the Korea Community Health Survey (KCHS), which is a nationally representative, cross-sectional survey that has been conducted by the Korea Centers for Disease Control and Prevention (KCDC) regularly since 2008 to gather regional data for the planning, monitoring, and evaluation of community health services. We used data from the survey that was conducted from 16 August to 31 October 2020. All participants provided informed consent, and all study protocols and procedures of the KCHS were reviewed and approved by the KCDC Institutional Review Board.

For community-level factors, the public data that confirmed COVID-19 patients by do-level were released by the KCDC. COVID-19 patient trends in South Korea were analyzed using the statistics of 16 August 2020, namely, “daily new confirmed patient count” and “cumulative number of patients”, to coincide with the commencement of the survey. In addition, we used data from 2020 to measure the financial independence ratios of local governments by do-level. The financial independence ratio of local governments was calculated according to the following formula: (Local tax + Non-tax revenue + Local shared tax)/General account budget. A higher level of financial independence facilitates greater budgetary discretion for community health [22].

### 2.1. Participants

In our study, we included only those participants in the KCHS in 2019 and 2020 who were diagnosed with diabetes (n = 53,313). Those who answered they do not know about the screening of diabetic complications were excluded (n = 418). Of these, any participants with missing data were excluded (n = 1424). Our study did not require approval from the institutional review board or informed consent because the KCHS is a secondary dataset available in the public domain and contains deidentified information.

### 2.2. Variables

The dependent variables were screening for diabetic retinopathy and nephropathy based on “yes” or “no” answers, using the following questions: “Have you ever had an eye examination to see if diabetic eye complications occurred during the past year?”, and “Have you ever had a precise urine test (microalbuminuria) to see if diabetic complications in the kidneys developed during the past year (with the exception of the stick test)?”

Individual-level variables included sociodemographic, socioeconomic, health status, and diabetes-related variables. Sociodemographic factors were age and sex (male, female). The socioeconomic factors included education level (middle school or lower, high school, and university or higher), marital status (yes or no), occupational categories (self-employed and employers, salary workers, unpaid family workers, inoccupation), and household income (high, middle high, middle low, and low). Diabetes-related variables included awareness of glucose levels (whether the patient was aware of their glucose level) and currently being treated for diabetes (yes or no). Health status variables were diagnosis of hypertension (yes or no) and depression (Have you felt sad or hopeless for more than two weeks in the past year?).

Community-level variables were based on region by level of COVID-19 spread. Regional cut-off points were divided into areas with a COVID-19 incidence of over 10 per 10,000 (very severe), 5 to 10 (severe), 1 to 5 (moderate), and 1 or less (mild). Additionally, the region size was categorized into three entities (metropolitan, city, and rural). Financial independence ratios of local regions were categorized into four quartiles, with Q1 and Q4 being the lowest and highest levels, respectively.

### 2.3. Statistical Analysis

The chi-squared test and *t*-test were used to determine significant differences in variables between participants who did and did not undergo fundus examinations or microalbuminuria tests. H_0_ assumes that there is no difference between the two groups in a particular variable. An observed effect as large or larger if H_0_ is true calculated a P value as the probability. The strength of evidence is estimated by the *P* value against H_0_. The strength of evidence is estimated by the *p*-value against H_0_. The larger the *p*-value, the weaker the evidence against H_0_ [23]. To investigate the effect of individual- and community-level variables on an individual’s likelihood of undergoing a fundus examination or microalbuminuria test, we performed two-level hierarchical models that assessed the relationship between fundus examination or microalbuminuria test and variables of both individual- and community-level factors.

We conducted a logistic regression model using PROC GLIMMIX to estimate a generalized hierarchical linear model since the outcome variable was binary and not normally distributed. We constructed four models. First, the null model was a two-level model of individuals nested within communities and did not include variables. This model showed a baseline for comparing the size of contextual variation in diabetes complications screening in subsequent models. Models 2 and 3 were the same as the null model but contained individual and community variables, respectively. Model 2 determined the effect of individual variables on diabetes complications screening, and Model 3 determined the effect of community variables. The final model (Model 4) included both individual- and community-level variables and estimated the net effect of community variables over the individual variables. Odds ratios (ORs) and 95% confidence intervals (CIs) were computed. The data were analyzed using SAS 9.4 (SAS Institute Inc.; Cary, CA, USA).

## 3. Results

The total population of the surveys in 2019 and 2020 included 458,123 individuals. We targeted individuals who were diagnosed with diabetes; therefore, we excluded individuals without diabetes (n = 404,810). A total 51,471 participants were finally selected for this study after excluding those with missing data (n = 1842).

Table 1 shows the total number of confirmed cases and the fatality rate of COVID-19 as of 16 August 2020. The incidence rate of COVID-19 was the highest in Daegu at 28.6 per 100,000 population. The incidence rate of COVID-19 was the lowest in Jeollanam-do Province at 0.24 per 100,000 population.

Table 2 presents the general characteristics of the study participants. Of the 51,471 participants diagnosed with diabetes, there were 25,336 (49.2%) men and 26,135 (50.8%) women. Before COVID-19, 12,315 (48.2%) people were tested for DN, and 10,360 (48.1%) people were tested for DR. After COVID-19, 12,467 (48.1%) people were tested for DN, and 10,674 (41.2%) people were tested for DR. However, there is no difference between the two groups in both DN (*p*-value 0.948) and DR (*p*-value 0.109).

The ORs for factors associated with DN screening were determined using multilevel logistic regression analysis and are shown in Table 3. The random effect covariance was 0.318 (standard error: 0.031) in the null model, and the intraclass correlation coefficient value was 0.0881, indicating that 8.81% of the variability in the screening was accounted for by communities. Model 4 shows both individual and community variables. The percentage change of variance was 37.7% ((0.318–0.198)/0.318 × 100) and the log likelihood ratio was 66,502.07, indicating that Model 4 was the best fitting model in this study. In Model 4, the OR of DN screening after COVID-19 (OR 0.97, 95% CI 0.93–1.00) was lower than that before COVID-19.

Table 4 shows the ORs for factors associated with DR screening that were determined using multilevel logistic regression analysis. The random effect covariance was 0.212 (standard error: 0.0212) in the null model, and the intraclass correlation coefficient value was 0.0605, indicating that 6.05% of the variability in the screening was accounted for by communities and that the odds of DR screening can be accounted for by communities. Model 4 shows both individual and community variables. The percentage change of variance was 43.3% ((0.212–0.118)/0.212 × 100) and the log likelihood ratio was 66,107.27, indicating that Model 4 was the best fitting model in this study. In Model 4, the OR of retinopathy examination after COVID-19 was not significant (OR 0.99, CI 0.96–1.03).

Table 5 presents the results of the subgroup analysis of the association between individual and community variables and the diabetic complications screenings. Regarding the community variables, the ORs of regions with “very severe” in terms of being affected by COVID-19 spread showed a negative association with both diabetes complications screenings after the outbreak of COVID-19 compared to before the outbreak of COVID-19 (DN: OR 0.71, 95% CI 0.57–0.88; DR: OR 0.74, 95% CI 0.59–0.92). However, the ORs of regions with “severe” in terms of being affected by COVID-19 spread showed an increased association with DN screening after the outbreak of COVID-19 compared to before the outbreak of COVID-19 (DN: OR 1.14, 95% CI 1.02–1.28).

## 4. Discussion

We investigated the differential effects on diabetes complications screening before and after COVID-19, and in particular, we investigated factors that affected diabetes complications screening in South Korea by using nationally representative survey data. Pre and post COVID-19 comparisons revealed a relationship with the number of screenings for DN complications in people with diabetes and the COVID-19 outbreak, while no relationship was found with the number of screenings for DR complications. Healthcare utilization decreased during the pandemic because of factors such as the social distancing policy and fears of contracting the virus within health facilities [24]. Internal medicine visits may have been more affected than ophthalmic visits by the recognition that the risk of exposure to COVID-19 was higher. Furthermore, compared with before COVID-19, our study found a negative association with screening for both diabetes complications among regions with “very severe” incidence of COVID-19 infection. However, the region where the spread of COVID-19 was “severe” had a positive association with the DN screening. The characteristics of the “severe” regions were rural, with a low population density of people who are older, and with a low income (Table 5).

Given the long-lasting outcome of the COVID-19 pandemic for the health care system, preventive care was affected to distancing policies or fears of COVID-19 infection [25,26]. However, diabetes complications are among the primary factors that increase medical costs and reduce the quality of life [27,28]. therefore this study suggests that the prevention and continuity of care for diabetic complications is very important in reducing economic burden and social cost [29,30]. Since our study was conducted in the early stages after the COVID-19 outbreak, it may have had a slight effect on screening for diabetic complications. However, as the COVID-19 pandemic continued, preventive care was more affected, and this may have led to more personal and national economic losses.

The COVID-19 pandemic is a unique situation requiring special measures such as social distancing to curb the spread of infection. The pandemic has had an enormous impact on society and healthcare [20]. However, it is unclear whether the decrease in healthcare use is in response to the implementation of social distancing or due to patients’ fears of COVID-19 infection [31]. Our study shows that diabetes complications screening has a significantly negative association with people who are being treated with medication for diabetes. The spread of COVID-19 may have curtailed essential medical care for patients with diabetes. If policies such as social distancing for limiting the spread of COVID-19 reduce necessary medical care, they may also impose additional costs. Screening of diabetic complications is important because diabetic complications increase medical costs, reduce quality of life, increase mortality, and increase social burdens [32,33]. For the early detection and appropriate treatment of diabetic complications, including DR and DN, annual screening scheduled by a physician is recommended for all patients with diabetes [34,35].

It is important to state that our study was conducted during the initial stage of the pandemic in South Korea using a nationally representative database to determine the association of diabetes complications screening and the outbreak of COVID-19. As COVID-19 is still ongoing, it will be important to know the current situation of non-COVID-19-related healthcare.

Our study has certain limitations. First, it was based on data from a cross-sectional study. Therefore, although associations could be confirmed, causality could not be evaluated. Second, our study relied on self-reported data. Future studies will also need to perform precise measurements of diabetes complications screening. Third, the question about diabetes complications screening was related to the previous year; however, the survey period was from August to October 2020. Therefore, the 2020 KCHS is not fully representative of the effects of COVID-19. Fourth, clinical information for each individual with regard to diabetes, including the duration of diabetes, HbA1c levels, blood pressure, and other eye disease morbidities, was not determined in this study due to the limited availability of such information from the data. Fifth, this study did not consider community interventions or policies. Finally, our findings can be explained by the timing of the survey because it was conducted between August and October 2020, when the second wave of COVID-19 was underway in South Korea. Screening for diabetes complications would also have been affected by the first wave of COVID-19. The first major outbreak in South Korea occurred from February to March 2020, when a large number of COVID-19 infections occurred in Daegu [15,25]. Therefore, we conducted this study reflecting only the situation in the early stages of COVID-19 in Korea. If more data are collected, further research will be needed.

## 5. Conclusions

The COVID-19 outbreak was associated with a reduction in the use of diabetes nephropathy complications screening. Additionally, only regions heavily affected by COVID-19 spread showed a negative association with diabetes complications screening compared to before the COVID-19 outbreak. In this regard, it appears that many patients were unable to attend outpatient care due to COVID-19. As such, these patients should be encouraged to visit clinics for diabetes complications screening. Furthermore, alternative methods need to be developed to support these patients. Through these efforts, the development of diabetes-related complications should be prevented, and the costs associated with these complications will be reduced.

## Figures and Tables

**Table 1 ijerph-19-05436-t001:** The total number of confirmed cases as of 16 August 2020 as well as case and fatality rates by region.

Region	Confirmed Cases	
	Cases ^a^	Deaths ^b^
	% ^a^	% ^b^
Total	2.7	2.2
Busan	0.6	1.4
Chungcheongbuk-do Province	0.5	0.0
Chungcheongnam-do Province	0.9	0.5
Daegu	28.6	2.7
Daejeon	1.1	1.2
Gwangju	1.5	0.9
Gangwon-do Province	0.5	3.7
Gyeonggi-do Province	1.4	1.7
Gyeongsangbuk-do Province	5.3	3.8
Gyeongsangnam-do Province	0.5	0.0
Incheon	1.4	0.7
Jejudo Island	0.4	0.0
Jeollabuk-do Province	0.2	0.0
Jeollanam-do Province	0.2	0.0
Sejong	1.5	0.0
Seoul	2.1	0.7
Ulsan	0.6	1.5

^a^ Incidence rate of COVID-19 per 100,000 population; ^b^ fatality rate of COVID-19.

**Table 2 ijerph-19-05436-t002:** General characteristics of the study population.

Variables	Diabetic Nephropathy Screening							Diabetic Retinopathy Screening				
	Total		Yes		No		*p*-Value	Yes		No		*p*-Value
	N	%	N	%	N	%		N	%	N	%	
	51,471	100.0	26,689	51.9	24,782	48.1		30,437	59.1	21,034	40.9	
**Individual level**												
**COVID-19 outbreak**												
Before COVID-19 (2019)	25,570	49.7	12,315	48.2	13,255	51.8	0.948	10,360	40.5	15,210	59.5	0.109
After COVID-19 (2020)	25,901	50.3	12,467	48.1	13,434	51.9		10,674	41.2	15,227	58.8	
**Sex**												
Men	25,336	49.2	12,525	49.4	12,811	50.6	<0.0001	10,251	40.5	15,085	59.5	0.065
Women	26,135	50.8	12,257	46.9	13,878	53.1		10,783	41.3	15,352	58.7	
**Age (years)**	51,471	100	65.94 (11.25)		67.55 (11.78)		<0.0001	66.25 (11.00)		67.14 (11.91)		<0.0001
**Marital status**												
Living w/spouse	35,143	68.3	17,529	49.9	17,614	50.1	<0.0001	14,791	42.1	20,352	57.9	<0.0001
Living w/o spouse	16,328	31.7	7253	44.4	9075	55.6		6243	38.2	10,085	61.8	
**Educational level**												
Middle school or less	30,298	58.9	13276	43.8	17,022	56.2	<0.0001	11,332	37.4	18,966	62.6	<0.0001
High school	13,456	26.1	7162	53.2	6294	46.8		6034	44.8	7422	55.2	
College or over	7717	15.0	4344	56.3	3373	43.7		3668	47.5	4049	52.5	
**Occupational categories**												
Self-employed and employers	10,852	21.1	5033	46.4	5819	53.6	<0.0001	4084	37.6	6768	62.4	<0.0001
Salary workers	12,028	23.4	5933	49.3	6095	50.7		4936	41.0	7092	59.0	
Unpaid family workers	2390	4.6	1038	43.4	1352	56.6		879	36.8	1511	63.2	
Inoccupation	26,201	50.9	12778	48.8	13,423	51.2		11,135	42.5	15,066	57.5	
**Household income**												
Low	12,589	24.5	5159	41.0	7430	59.0	<0.0001	4393	34.9	8196	65.1	<0.0001
Mid-low	13,205	25.7	6153	46.6	7052	53.4		5303	40.2	7902	59.8	
Mid-high	12,516	24.3	6396	51.1	6120	48.9		5417	43.3	7099	56.7	
High	13,161	25.6	7074	53.7	6087	46.3		5921	45.0	7240	55.0	
**Recognition of own glucose level**												
Yes	36,826	71.5	19,380	52.6	17446	47.4		16,607	45.1	20,219	54.9	<0.0001
No	14,645	28.5	5402	36.9	9243	63.1		4427	30.2	10,218	69.8	
**Medication of DM**												
Yes	48,138	93.5	23,828	49.5	24,310	50.5	<0.0001	20,278	42.1	27,860	57.9	<0.0001
No	3333	6.5	954	28.6	2379	71.4		756	22.7	2577	77.3	
**Diagnosis of hypertension**												
Yes	31,879	61.9	15,369	48.2	16,510	51.8	0.716	12,929	40.6	18,950	59.4	0.069
No	19,592	38.1	9413	48.0	10,179	52.0		8105	41.4	11,487	58.6	
**Depression**												
Yes	3812	7.4	1937	50.8	1875	49.2	<0.0001	1696	44.5	2116	55.5	<0.0001
No	47,659	92.6	22,845	47.9	24,814	52.1		19,338	40.6	28,321	59.4	
**Community level**												
**Region by level of COVID-19 spread**												
Very severe	1413	2.7	696	49.3	717	50.7	<0.0001	572	40.5	841	59.5	<0.0001
Severe	5618	10.9	2144	38.2	3474	61.8		1989	35.4	3629	64.6	
Moderate	15,802	30.7	9194	58.2	6608	41.8		7669	48.5	8133	51.5	
Mild	28,638	55.6	12,748	44.5	15,890	55.5		10,804	37.7	17,834	62.3	
**Region**												
Metropolitan	4168	8.1	2677	64.2	1491	35.8	<0.0001	2348	56.3	1820	43.7	<0.0001
City	17,078	33.2	9199	53.9	7879	46.1		7661	44.9	9417	55.1	
Rural	30,225	58.7	12,906	42.7	17,319	57.3		11,025	36.5	19,200	63.5	
**Financial independence ratio**												
Q1	8720	16.9	3369	38.6	5351	61.4	<0.0001	2731	31.3	5989	68.7	<0.0001
Q2	13,834	26.9	5893	42.6	7941	57.4		5217	37.7	8617	62.3	
Q3	14,816	28.8	7301	49.3	7515	50.7		6223	42.0	8593	58.0	
Q4	14,101	27.4	8219	58.3	5882	41.7		6863	48.7	7238	51.3	

The *p*-values reflect *t*-tests for continuous variables and χ^2^ tests for dichotomous/categorical variables.

**Table 3 ijerph-19-05436-t003:** Adjusted odds ratios of diabetes nephropathy screening by characteristics of an individual- and area-level, multilevel model.

Variables	Diabetes Nephropathy Screening															
	Model 1				Model 2				Model 3				Model 4			
	OR	95% CI			OR	95% CI			OR	95% CI			OR	95% CI		
**Individual level** **(fixed effect)**																
**COVID-19 outbreak**																
Before COVID-19 (2019)					1.00								1.00			
After COVID-19 (2020)					0.97	(0.93–1.00)							0.97	(0.93–1.00)		
**Sex**																
Men					1.00								1.00			
Women					1.04	(1.00–1.09)							1.05	(1.01–1.10)		
**Age (years)**					1.00	(0.99–1.00)							1.00	(0.99–1.00)		
**Marital status**																
Living w/spouse					1.00								1.00			
Living w/o spouse					0.89	(0.85–0.93)							0.88	(0.85–0.92)		
**Educational level**																
Middle school or less					0.78	(0.73–0.83)							0.78	(0.74–0.84)		
High school					0.92	(0.87–0.98)							0.93	(0.87–0.98)		
College or over					1.00								1.00			
**Occupational** **categories**																
Self-employed and employers					0.88	(0.83–0.93)							0.90	(0.85–0.95)		
Salary workers					0.84	(0.80–0.89)							0.86	(0.81–0.90)		
Unpaid family workers					0.91	(0.83–1.00)							0.93	(0.85–1.02)		
Inoccupation					1.00								1.00			
**Household income**																
Low					0.88	(0.83–0.94)							0.89	(0.83–0.95)		
Mid-low					0.96	(0.90–1.01)							0.96	(0.91–1.02)		
Mid-high					1.00	(0.95–1.05)							1.00	(0.95–1.05)		
High					1.00								1.00			
**Recognition of own glucose level**																
Yes					1.66	(1.59–1.73)							1.66	(1.59–1.73)		
No					1.00								1.00			
**Medication of DM**																
Yes					2.81	(2.59–3.05)							2.86	(2.63–3.10)		
No					1.00								1.00			
**Diagnosis of hypertension**																
Yes					1.11	(1.06–1.15)							1.10	(1.06–1.15)		
No					1.00								1.00			
**Depression**																
Yes					1.12	(1.04–1.20)							1.12	(1.04–1.20)		
No					1.00								1.00			
**Community level** **(random effect)**																
**Region by level of COVID-19 spread**																
Very severe									1.01	(0.69–1.48)			0.98	(0.67–1.44)		
Severe									0.70	(0.55–0.89)			0.76	(0.60–0.97)		
Moderate									1.39	(0.98–1.99)			1.44	(1.01–2.05)		
Mild													1.00			
**Region**																
Metropolitan									1.00				1.00			
City									0.71	(0.56–0.88)			0.71	(0.57–0.89)		
Rural									0.67	(0.48–0.95)			0.70	(0.50–0.99)		
**Financial independence ratio**																
Q1									0.69	(0.47–1.01)			0.79	(0.53–1.16)		
Q2									0.97	(0.66–1.42)			0.98	(0.66–1.43)		
Q3									1.02	(0.74–1.41)			1.09	(0.79–1.49)		
Q4									1.00				1.00			
**Between-area variance (SE)**	0.318 (0.031) *				0.290 (0.028) *				0.200 (0.020) *				0.198 (0.0020) *			
**Model fitness**																
−2 log likelihood	68,347.24				66,595.32				68,238.72				66,502.07			
AIC	68,351.24				66,631.32				68,258.72				66,554.07			
**Intraclass correlation coefficient (%) ^a^**	8.81 ^a^															

adj. OR, adjusted odds ratio; CI, confidence interval; IACI; industrial accident compensation insurance; SE, standard error; AIC, Akaike information criterion. ^a^ A total of 8.81% of the variability in the odds of suicidal ideation is accounted for by the areas in the study. * *p* < 0.0001.

**Table 4 ijerph-19-05436-t004:** Adjusted odds ratios of diabetes retinopathy screening by characteristics of an individual- and area-level, multilevel model.

Variables	Diabetic Retinopathy Screening															
	Model 1				Model 2				Model 3				Model 4			
	OR	95% CI			OR	95% CI			OR	95% CI			OR	95% CI		
**Individual level** **(fixed effect)**																
**COVID-19 outbreak**																
Before COVID-19 (2019)					1.00								1.00			
After COVID-19 (2020)					0.99	(0.95–1.02)							0.99	(0.96–1.03)		
**Sex**																
Men					1.00								1.00			
Women					1.20	(1.15–1.26)							1.21	(1.16–1.26)		
**Age (years)**					1.00	1.00							1.00	1.00		
**Marital status**																
Living w/spouse					1.00								1.00			
Living w/o spouse					0.90	(0.86–0.94)							0.90	(0.86–0.94)		
**Educational level**																
Middle school or less					0.75	(0.70–0.80)							0.74	(0.69–0.79)		
High school					0.91	(0.85–0.96)							0.90	(0.85–0.96)		
College or over					1.00								1.00			
**Occupational categories**																
Self-employed and employers					0.84	(0.80–0.89)							0.87	(0.82–0.91)		
Salary workers					0.84	(0.80–0.88)							0.85	(0.81–0.89)		
Unpaid family workers					0.85	(0.77–0.93)							0.90	(0.82–0.99)		
Inoccupation					1.00								1.00			
**Household income**																
Low					0.83	(0.78–0.89)							0.85	(0.79–0.90)		
Mid-low					0.96	(0.90–1.01)							0.97	(0.91–1.02)		
Mid-high					0.99	(0.93–1.04)							1.00	(0.94–1.05)		
High					1.00								1.00			
**Recognition of own glucose level**																
Yes					1.67	(1.60–1.75)							1.69	(1.62–1.77)		
No					1.00								1.00			
**Medication of DM**																
Yes					2.73	(2.50–2.98)							2.78	(2.55–3.03)		
No					1.00								1.00			
**Diagnosis of hypertension**																
Yes					1.02	(0.98–1.06)							1.02	(0.98–1.06)		
No					1.00								1.00			
**Depression**																
Yes					1.17	(1.09–1.25)							1.17	(1.09–1.25)		
No					1.00								1.00			
**Community level** **(random effect)**																
**Region by level of COVID-19 spread**																
Very severe									0.89	(0.65–1.23)			0.88	(0.65–1.20)		
Severe									0.83	(0.68–1.01)			0.87	(0.72–1.05)		
Moderate									1.17	(0.88–1.58)			1.20	(0.90–1.59)		
Mild									1.00				1.00			
**Region**																
Metropolitan									1.00				1.00			
City									0.65	(0.54–0.79)			0.66	(0.55–0.79)		
Rural									0.57	(0.43–0.76)			0.62	(0.48–0.82)		
**Financial independence ratio**																
Q1									0.71	(0.51–0.97)			0.76	(0.56–1.03)		
Q2									1.04	(0.76–1.43)			1.03	(0.76–1.41)		
Q3									1.06	(0.81–1.38)			1.09	(0.84–1.40)		
Q4									1.00				1.00			
**Between-area variance (SE)**	0.212 (0.0212) *				0.179 (0.0183) *				0.128 (0.0136) *				0.118 (0.0128) *			
**Model fitness**																
−2 log likelihood	67,829.8				66,111.55				67,717.07				66,107.27			
AIC	67,833.80				66,151.55				67,737.07				66,159.27			
**Intraclass correlation coefficient (% ) ^a^**	6.05 ^a^															

adj. OR, adjusted odds ratio; CI, confidence interval; IACI, industrial accident compensation insurance; SE, standard error; AIC, Akaike information criterion. ^a^ A total of 6.05% of the variability in the odds of suicidal ideation is accounted for by the areas in the study. * *p* < 0.0001.

**Table 5 ijerph-19-05436-t005:** The results of subgroup analysis stratified by independent variables.

Variables	Diabetic Nephropathy Screening			Diabetic Retinopathy Screening		
	Before COVID-19	After COVID-19		Before COVID-19	After COVID-19	
	OR	OR	95% CI	OR	OR	95% CI
**Age**						
19–39	1.00	0.84	(0.64–1.12)	1.00	0.82	(0.62–1.10)
40–49	1.00	0.84	(0.72–0.97)	1.00	0.99	(0.85–1.16)
50–59	1.00	0.86	(0.79–0.94)	1.00	1.01	(0.93–1.10)
60–69	1.00	1.05	(0.99–1.13)			
70–	1.00	0.97	(0.92–1.03)	1.00	0.98	(0.94–1.02)
**Educational level**						
Middle school or less	1.00	0.98	(0.93–1.03)	1.00	1.01	(0.96–1.06)
High school	1.00	0.92	(0.86–0.99)	1.00	0.97	(0.90–1.04)
College or over	1.00	0.97	(0.88–1.07)	1.00	0.99	(0.90–1.09)
**Household income**						
Low	1.00	1.02	(0.95–1.10)	1.00	1.06	(0.98–1.14)
Mid-low	1.00	0.96	(0.89–1.03)	1.00	0.92	(0.85–0.99)
Mid-high	1.00	0.99	(0.92–1.07)	1.00	1.02	(0.94–1.10)
High	1.00	0.88	(0.82–0.95)	1.00	0.96	(0.91–1.05)
**Community level**						
**Region by level of COVID-19 spread**						
Very severe	1.00	0.71	(0.57–0.88)	1.00	0.74	(0.59–0.92)
Severe	1.00	1.14	(1.02–1.28)	1.00	1.01	(0.90–1.12)
Moderate	1.00	0.99	(0.93–1.06)	1.00	1.00	(0.94–1.07)
Mild	1.00	0.94	(0.89–0.99)	1.00	1.01	(0.96–1.06)
**Region**						
Metropolitan	1.00	0.96	(0.84–1.11)	1.00	1.10	(0.96–1.26)
City	1.00	0.88	(0.82–0.94)	1.00	0.88	(0.83–0.94)
Rural	1.00	1.02	(0.98–1.07)	1.00	1.05	(1.00–1.11)

## Data Availability

The data that support the findings of this study are openly available in the Korea Community Health Survey at (https://chs.kdca.go.kr, accessed on 20 August 2021).

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
