# Peer review of "Screening for Diabetes Complications during the COVID-19 Outbreak in South Korea"

_ijerph, 2022, doi:10.3390/ijerph19095436_

Round 1
Reviewer 1 Report
In this article the authors describe a decrease in screening appointments for complication in those with diabetes. The article describes a complex modelling strategy that accounts for community and individual level factors that could affect diabetes complication screening. The article is well written and describes the research question and methods used in good detail. However, I have reservations about the interpretation of results and discussion of the findings that I believe need to be addressed before publication.
Major points
- The results demonstrate no evidence of an association between COVID-19 pandemic and diabetic retinopathy screening, but the discussion references “a reduction in the use of diabetes complications screening.” This should specifically refer to diabetes nephropathy screening only given the results in tables 3 and 4.
- Further analysis (or discussion) should be added about the public health implications of this reduction in diabetes nephropathy screening since the relative effect size is small, what is the absolute effect on people with diabetes?
- At line 177, the results state that “regions heavily affected by COVID-19 spread showed a negative association with diabetes complication screening” however this should be explicit that this is true only for the “very severe” category in Table 5, and discussion should be added as to why only this most severe category shows an effect and why the “severe” category shows an increase following the pandemic in DN screening. In addition, how sensitive are these findings to your chosen cut-offs for “very severe” etc.?
- From lines 152-155, the interpretation of these other effects is susceptible to the Table 2 fallacy and should be removed or discussion of this limitation added.
- The interpretation of Table 1 is confusing. Percentages are presented in the table, but rates are described in the text. 258.9 cases per 100,000 people is not the same as 28.6% of the population as shown in the table. This needs to be consistent and described more clearly.
Minor points
- Line 82, why was a complete case analysis chosen? This needs justification.
- Line 92, why was age categorised and not used as a continuous variable?
- Lines 91-100, since you removed any missing data, it is important to show which variables had the most missing data somewhere in the report.
- Line 110, remove mention of statistically significant boundaries and describe the strength of evidence from p-values instead.
- Line 134, add the lowest incidence rate region to demonstrate the range of COVID-19 infection rates.
- How is Table 1 sorted? Presentation could be improved by sorting alphabetically or by case rates.
- Line 175, did the authors have sufficient power to conduct this many subgroup analyses? And if so, what is the risk from multiple hypothesis testing? Finally, if conducting a subgroup analysis, what is the rationale for the effect of COVID-19 being different on diabetes complications screening between these subgroups? I am not sure it is sufficient to repeat the analysis for all subgroups without justification.
Reviewer 2 Report
This study compares the rate of diabetes complications screenings in South Korea before and during COVID-19 pandemic. The authors find a decrease in the rate of both diabetic retinopathy and diabetic nephropathy in 2020, compared to the previous year. The authors then decompose these effects based on an extensive set of patient and region characteristics.
This is an interesting paper that contributes to the literature of COVID-19 impact on outpatient medical services in general and diagnostic screenings in particular. The study design prevents the causal identification of the pandemic's impact on the provision of care, but this issue is common across nearly all papers dealing with the impact of the pandemic. The correlational analysis is extensive and clear.
Major comments:
- Reference to the existing literature on the impact of the pandemic on the provision of services should appear in the introduction. The omission of a discussion of this literature in the introduction is glaring, particularly since the authors write that this is the first study on the impact of the pandemic on diabetes complications screenings. While that may be true, other studies have discussed disruptions to other types of screenings. I note that the authors do discuss this literature in the discussion.
- Table 1: the authors should include a column that details the total number of individuals residing in the region (the denominator for the case %). The first row ("Confirmed Cases") is redundant. The percent terms seem to be two orders of magnitude off. For example, the total death rate should be 0.0217%, not 21.7%. The "%" columns should indicate what is the denominator (% of cases, for the death rate, or simply CFR).
- Table 2: The "TOTAL" row under "Diabetic Retinopathy Screening" is redundant, you can place the first "TOTAL" row outside the "Diabetic Nephropathy Screening" header and have it be a denominator for both types of screenings. The "No" columns are redundant as well. You should describe what test is performed when discussing P-value (not just the test statistic, but what is tested).
- P. 8 Lines 181-182: You describe regional household income, but this variable does not appear in the actual table. It should either be included, or the discussion of this result be omitted. Please make sure references to variables that are not used in the table do not appear in the text.
Round 2
Reviewer 1 Report
Please check the attachment.

Author Response
Revision Note for MS ID: JCH-21-0409
Screening for diabetes complications during the COVID-19 outbreak in South Korea
(ijerph-1581328)
In this article the authors describe a decrease in screening appointments for complication in those with diabetes. The article describes a complex modelling strategy that accounts for community and individual level factors that could affect diabetes complication screening. The article is well written and describes the research question and methods used in good detail. However, I have reservations about the interpretation of results and discussion of the findings that I believe need to be addressed before publication.
Response to Reviewer #1’s comments
Major concern 1: The results demonstrate no evidence of an association between COVID-19 pandemic and diabetic retinopathy screening, but the discussion references “a reduction in the use of diabetes complications screening.” This should specifically refer to diabetes nephropathy screening only given the results in tables 3 and 4.
Re: The revised section in the discussion is an improvement since it explicitly mentions and interprets the findings in relation to diabetes complication screening. However, is it important to mention that the same association was not found for diabetes retinopathy screening in your analysis
Response 1: Thank you for your great effort in reviewing our manuscript and giving meaningful feedback. As you mentioned above, we added the explanation of our discussion
Revised manuscript, line 198~204, page 5: Compared to 2019 (pre-COVID-19), there’s relationship with the number of screenings for diabetes nephropathy complications in people with diabetes and COVID-19 outbreak. However, there’s no relationship with the number of screenings for diabetes retinopathy complications in people with diabetes. It can be observed that healthcare utilization was decreased during pandemics because of such factors as social distancing policy and fears of contracting the virus within health facilities. [24]
Major concern 2: Further analysis (or discussion) should be added about the public health implications of this reduction in diabetes nephropathy screening since the relative effect size is small, what is the absolute effect on people with diabetes?
Re: The new paragraph (at line 227) is an excellent addition
Major concern 3: At line 177, the results state that “regions heavily affected by COVID-19 spread showed a negative association with diabetes complication screening” however this should be explicit that this is true only for the “very severe” category in Table 5, and discussion should be added as to why only this most severe category shows an effect and why the “severe” category shows an increase following the pandemic in DN screening. In addition, how sensitive are these findings to your chosen cut-offs for “very severe” etc.?
Re: I do not think this concern has been addressed sufficiently. The revised section does not mention the terms “severe” or “very severe” and this needs to be included in the discussion section and in the results when interpreting Table 5. Also, the paragraph concerning Table 5 (starting at line 197) is confusing as it mentions ORs from both Table 5 and Table 4. There needs to be a separate paragraph for Table 5 that mentions the differences between the observed association between very severe/severe COVID-19 spread, and DN screening. The reason for this then needs to be in the discussion. The manuscript currently does not provide a sufficient discussion of why a “severe” COVID-19 spread resulted in an increase in the odds of DN screening in Table 5. In addition, there is still no discussion or sensitivity analysis about the definition of “severe” and “very-severe” COVID-19 spread. If you change these boundaries, does this affect your findings?
Response 1: Thank you for your comment. As you mentioned above, our result is explicit that this is true only for the “very severe” category. So we describe the explanation of our discussion and result exactly. Also, we add the explanation about why a “severe” COVID-19 spread resulted in an increase in the odds of DN screening in Table 5. For supplementary explanation, variables from previous subgroups were added to the table 5.
Revised manuscript, line 185~192, page 4: Regarding the community variables, the ORs of regions with “very severe” affected by COVID-19 spread showed a negative association with both diabetes complications screening after the outbreak of COVID-19 compared to before the outbreak of COVID-19 (DN: OR 0.71, 95% CI 0.57-0.88; DR: OR 0.74, 95% CI 0.59-0.92). But the ORs of regions with “severe” affected by COVID-19 spread showed an increased association with DN screening after the outbreak of COVID-19 compared to before the outbreak of COVID-19 (DN: OR 1.14, 95% CI 1.02-1.28)
Revised manuscript, line 204~211, page 5: Also, compared to before COVID-19, our study found that negative association with both diabetes complications screening was founded among region with “very severe” incidence of COVID-19 infection. However, the region where the spread of COVID-19 was severe had a positive association with the DN screening. The characteristics of the region in “severe” region are rural and aging. In table 5, people who are older and live in rural, have low income were not negative associated with DN screening. Further, it may have been less affected by COVID-19, due to the area with a low population density.
Major concern 4: From lines 152-155, the interpretation of these other effects is susceptible to the Table 2 fallacy and should be removed or discussion of this limitation added.
Re: This revision does not address the comment. The comment is in reference to the interpretation of the other effects in Table 3 (e.g. “The ORs increased in older groups…”). These variables are not the focus of this analysis, and their estimates should not be interpreted in the same way as the primary exposure (COVID-19 outbreak). Apologies, my comment may have been confusing as I was referring to the “the Table 2 fallacy” in epidemiological studies, not Table 2 in this manuscript. More information on the Table 2 fallacy is here https://doi.org/10.1093/aje/kws412.
Response 1: Thank you for your comment. we understood your comments. So we removed the mention that can be susceptible to the table2 fallacy. Thank you for your meaningful comments.
Major concern 5: The interpretation of Table 1 is confusing. Percentages are presented in the table, but rates are described in the text. 258.9 cases per 100,000 people is not the same as 28.6% of the population as shown in the table. This needs to be consistent and described more clearly.
Re: This comment has been addressed well.
Minor concern 2 Line 92, why was age categorized and not used as a continuous variable?
Re: These minor points have all been addressed, except minorpoint2. I would suggest that age should still be treated as a continuous and not a categorical variable., but if it is to be categorized then the authors should include an explanation of why they categorized it and why they chose the categories used in the analysis.
Response 1: Thank you for your comment. As you mentioned above, we change the age as a continuous variable. So we have revised all results of tables. Please see our revised manuscript Thank you for your meaningful comments.
Revised manuscript, line 113, page 3: age
Revised manuscript, line 129, page 3: The Chi-squared test and t-test was used to determine the significant differences in variables between participants who did and did not undergo fundus examinations or microalbuminuria tests.
Additional revision:
- To clarify our study, we revised the abstract and conclusion entirely. Please see our revised manuscript.
- We have revised all results of tables. Please see our revised manuscript
- we added the limitation.
Finally, our findings can be explained by the timing of the survey because it was con-ducted between August and October 2020, when the second wave of COVID-19 was underway in South Korea. Screening for diabetes complications would also have been affected by the first wave of COVID-19. The first major outbreak in South Korea occurred from February to March 2020, when a large number of COVID-19 infections occurred in Daegu.[15, 25]. So we conducted this study reflecting only the situation in the early stages of COVID-19 in Korea. If data are collected more, further research will be needed.
- if this manuscript \is confirmed, we want to receive the professional scientific editing service due to grammatical error.
